# Blockchain technology in healthcare: A systematic review

**Huma Saeed**[1], **Hassaan Malik** [1,2]*, **Umair Bashir**[1], **Aiesha Ahmad**[1], **Shafia Riaz**[1], **Maheen Ilyas**[1], **Wajahat Anwaar Bukhari**[1], **Muhammad Imran Ali Khan**[1]

**1** Department of Computer Science, National College of Business Administration & Economics Lahore, Multan, Pakistan, **2** Department of Computer Science, University of Management and Technology, Lahore, Pakistan

* f2019288004@umt.edu.pk

**Data Availability Statement:** All relevant data are within the manuscript and its Supporting information files.

**Funding:** The author(s) received no specific funding for this work.

## Abstract

Blockchain technology (BCT) has emerged in the last decade and added a lot of interest in the healthcare sector. The purpose of this systematic literature review (SLR) is to explore the potential paradigm shift in healthcare utilizing BCT. The study is compiled by reviewing research articles published in nine well-reputed venues such as IEEE Xplore, ACM Digital Library, Springs Link, Scopus, Taylor & Francis, Science Direct, PsycINFO, Ovid Medline, and MDPI between January 2016 to August 2021. A total of 1,192 research studies were identified out of which 51 articles were selected based on inclusion criteria for this SLR that presents the modern information on the recent implications and gaps in the use of BCT for enhancing the healthcare procedures. According to the outcomes, BCT is being applied to design the novel and advanced interventions to enrich the current protocol of managing, distributing, and processing clinical records and personal medical information. BCT is enduring the conceptual development in the healthcare domain, where it has summed up the substantial elements through better and enhanced efficiency, technological innovation, access control, data privacy, and security. A framework is developed to address the probable field where future researchers can add considerable value, such as data protection, system architecture, and regulatory compliance. Finally, this SLR concludes that the upcoming research can support the pervasive implementation of BCT to address the critical dilemmas related to health diagnostics, enhancing the patient healthcare process in remote monitoring or emergencies, data integrity, and avoiding fraud.

## Introduction

Healthcare is a system that includes 3 main components: (i) Main suppliers of services for medical treatment, For instance, doctors, nurses, technicians, and hospital administrations (ii) Emergency related services [1–4], and (iii) Health and health-oriented service users, specific patients. In the current study, to encourage, preserve or restore the health of beneficiaries, we examine the health maintenance to include technology-based remotely controlling services increased by constituent service providers [5–9]. In the medical field, every year, there are

**Competing interests:** The authors have declared that no competing interests exist.

more security and privacy breaches, in 2017, more than 300 breaches were reported, and up to 37 million records were affected during 2010–2017 [10, 11]. The growing digitization of medical care has advanced the acknowledgment of issues about secure storage, accessing of patients' medical records, ownership, and medical data from associated sources [12–16]. Blockchain is recommended as a method of addressing critical issues faced by healthcare, for instance, protected sharing of health records and adherence to data privacy laws [17–19].

Blockchain is a particular type of database that can be managed by the network of authenticated members or nodes [13] and stores immutable information blocks that can be strongly exchanged without interference by third parties [10]. With cryptographic signatures and the use of consensus algorithms which are implemented as key enablers in their application, data is stored and registered [20]. The capability of preserving data is a major aim for using the BCT particularly in healthcare [21], which is subject to massive sharing and dissemination of a significant amount of data [7]. In different stages, the development of blockchain technology, as well as its application in various contexts, had been materialized. The first phase of blockchain development was focused on cryptocurrencies, while the second focused on the use of smart contracts in industries like real estate and finance [11, 22]. The 3rd generation of evolution concentrated on employing blockchain in non-financial areas including government, culture [23], and healthcare space [22, 24]. Also, powered by revolutionary technical features such as data immutability [25], with the introduction of artificial intelligence, blockchain technology is having its 4th generation of evolution [26]. This asserted diversity in Blockchain's application spectrum can be attributed to its ability to build decentralized [27] and trustless transaction environments [28]. As blockchain can tackle serious issues, such as automated claim authentication [9] and public health management [29], the healthcare sector is a prime choice for the application of blockchain technology [30–32]. This technology allows patients to keep personal data and determine with whom this can be shared, thus resolving current data ownership, and sharing issues [28, 33]. At the same time, it allows recorded data to be integrated, modified, shared safely, and retrieved on time by relevant authorities using consensus protocols [31]. This is a significant benefit of the use of this technology in the healthcare system, as existing procedures need third parties to store the data [10]. Finally, because of possible human error, blockchain could potentially add accountability to data management processes [34] further decreasing the risks of mishandling or misusing recorded data [31]. Given the optimistic connotations of the effects of blockchain on social and business change, in contrast to previously defined expectations, it appears to be a discussion regarding its basic and derived advantages. A recent study indicates that while organizations will make substantial investments in the future in adopting blockchain-based technology due to a widespread perception that the advantages could be over-hype, they will probably accept a cautiously pragmatic approach [35]. It can be said that this technology has yet to fulfill its expectations [36], a fact that can be due to the prevalent adoption of block chain, particularly about regulatory barriers, to certain challenges [31]. The general public and specific users, for instance, patients or physicians are not acquainted with the way blockchain works, the technological features, or its advantages for data processing is another significant obstacle in promulgating the implementation of blockchain [35]. Suggest that it may take a considerable time for this technology to establish all anticipated stages of business transformation mainly because of the organizational, social, and implementation challenges, for example, security issues or governance reasons [22, 31]. This could also be exacerbated by general confusion regarding the use of blockchain regarding legal enforcement and regulations of the government. Current research focuses on supporting blockchain operational growth and speed-up its prevalence by overcoming these barriers.

However, previous studies have made little attempt to comprehensively summarize the existing knowledge by using SLRs [9–13]. For example, bibliometric techniques were used by [10] to provide a summary of blockchain research patterns and components related to the implementation of blockchain in the field of healthcare. In [9] the different blockchain platforms have been developed to deploy blockchain in healthcare. The study [11] addressed different examples of the implementation in the healthcare of blockchain technology, the problems, and their potential solutions. In diverse contexts where this technology was implemented [12], addressed design choices and tradeoffs made by the researchers. The research studies of [13, 14] have discussed the Blockchain-based applications throughout numerous industries and addressed many contexts of use for this technology in a broad manner. Recently [14] reviewed 39 studies to present an overview on common channels and other areas where blockchain technology is utilized for healthcare enhancement. Although these systematic literature reviews have a contribution to the extent of knowledge, their emphasis has been mainly on synthesizing or delineating blockchain technology patterns and areas [10, 11, 13, 14, 16]. However, researchers will get benefit from a concentrated discussion on the implications of its adoption [15], along with concrete obstacles and areas for progress for advancing the field, due to the reach and diversity of previous blockchain studies [11]. Through assimilating existing information and describing focus areas that require considerable academic attention, review-based research will assist in meeting these needs [11, 16–19]. As a result of this necessity, we perform an SLR on the blockchain technology application. This SLR presents a valuable overview of ongoing research, gaps in current knowledge, and future avenues of research as well. The contribution of this study is in two ways, this research adds to the emerging blockchain literature in healthcare. First regarding their implementation areas, restrictions, and recommendations, it offers an advanced and thematically ordered classification of previous literature. Second, we propose a synthesizing process according to the results of the SLR to detail possible topics that need academic attention to further update the existing body of literature.

The present study is organized as follows: In Section 2, we provide a thorough description of the research method utilized to search, screen, and select the literature. In Section 3, we present relevant review works that have been conducted in the field of health care using blockchain technology and discuss all the papers that have been selected, focusing on their main findings, and highlighting research gaps for future research. Finally, in Section 4we conclude this study.

## Methodology

SLRs always provide a thorough understanding of literature as it presents a complete and systematized review meeting all standard protocols in it [18, 37–39]. SLRs also help in the understanding of current information gaps and, as a result, the discovery of potential research avenues [19].

### Research questions

We conduct this SLR by addressing the following research questions (RQs).

**RQ1**: What is the advanced profile used for the employment of blockchain in the healthcare domain?
The purpose of this research question is to identify the number of research papers issued every year, the average citation received on research papers yearly, and academic contribution on the subject by Journals, publishing houses, and community.

**RQ2**: What are the major healthcare domains where blockchain technology has been implemented?
The purpose of this question is to identify the contexts in which blockchain technology has shown significant outcomes in healthcare.

**RQ3**: What are the existing problems and constraints raised by the previous studies in the healthcare field using blockchain technology?
The motivation behind this question is to identify existing problems and issues of blockchain technology in the healthcare field based on results, limitations, and conclusions of previous research studies.

**RQ4**: What are the potential healthcare avenues that would benefit from blockchain technology implementation?
The purpose of this question is to identify growing gaps and prospects of the future research agenda

## Research objectives

The research objectives (ROs) of the article herein presented are the following:

**RO1**: Establishing an archive of work that relates a wide topic about Blockchain in healthcare and offers an open dataset about Blockchain for all other researchers.

**RO2**: Identify a more focused set of studies that have used blockchain technology in healthcare applications.

**RO3**: Identify problems and constraints discussed in the healthcare field using blockchain technology.

**RO4**: Characterize existing solutions in the field of blockchain in healthcare and clarify the similarities and differences between them using a characterization framework.

## Research strategy

Nine databases—IEEE Xplore, ACM Digital Library, Springs Link, Scopus, Taylor & Francis, Science Direct, PsycINFO, Ovid Medline, MDPI—are recognized by previous studies as standard data sources of research papers about health informatics [40]. Reviewed papers have been outlined for understanding the research status of applying blockchain in health care. For the right database search the three keyword combinations existed as—"blockchain and Healthcare", or "medical Health" or Medical Management or Health Management. The above keywords were extracted from an article of previous literature i.e. SLRs) using similar keywords such as blockchain and medical healthcare.

## Study selection

The selection process aimed to find the articles that are the most relevant to the objective of this SLR. If there was the same paper in more than one source, as per our research, it was considered only once. The content of the papers chosen for the final sample was evaluated [39, 41] to make sure that the findings of the present SLR produced clear results and that is not biased. For reaching a consensus of final inclusion or exclusion, two of the researchers finalized the evaluation. After completing this, the discrepancies of individual assessments were addressed through discussion. A third author was engaged in analysis and debate in situations where the two writers did not find consensus. After the papers were found, the first move was to delete

redundant titles and those which are not connected in scrutiny. The standards for inclusion were limited to the hunt for String, and a study conducted by at least one of the following criteria for exclusion (EC) is omitted:

**Inclusion criteria (IC's).**

**IC1**: Studies are released any time on or before August 2021.

**IC2**: Studies are limited to the journal, conference, report, workshop, and symposium articles only.

**IC3**: Availability of complete texts in digital databases.

**IC4**: Proposed models or frameworks present.

**Exclusion criteria (EC).**

**EC1**: Exclude duplicated studies.

**EC2**: Eliminate preview, book chapters, magazines, thesis, monographs, and interview-based articles.

**EC3**: Exclude studies based on quality evaluation criteria.

**EC4**: Studies written in a language other than English.

The choice of papers was based on clear above discussed criteria for inclusion and exclusion. Below Fig 1 has been developed through the aspiration from the PRISMA diagram [42]. Fig 1 shows the study selection process.

## Results and discussion

This section describes the outcomes related to the systematic study RQs discussed above. 51 research studies have been selected to illustrate the outcomes of each RQs. Publication and other selection biases are a potential threat to validity in all SLR and we cannot exclude the possibility that some research studies were missed resulting in reduced precision and the potential for bias. Therefore, we made significant efforts in finding all eligible research articles, and conference proceedings from different well-reputed databases and by contacting experts in the BCT area through social media platforms. We believe that our work provides a significant contribution to the role of blockchain technology in health care.

### Selection results

Our search identified 1177 records, of which 1126 were screened as shown in Fig 1. 51 research articles were included in this SLR. The list of selected papers with descriptions of the overall classification results are discussed below.

**RQ1: What is the advanced profile used for the employment of blockchain in the healthcare domain?.** This SLR addresses the achieved descriptive records about the number of articles that have been published each year, publication source, the average citation received on research papers yearly (see Table 1). To complete this SLR, we have examined published surveys, systematic literature review (SLR), systematic reviews (SR), and research papers related to blockchain in healthcare, and published in the field of blockchain from 2018 to 2021. The number of highest citation research articles with the most citations is shown in Table 1.

Fig 2 demonstrates the number of articles published each year from 2018–2021. The four obvious outliers are existing from 2018 to 2021. In 2018, 24 articles were published, 16 articles

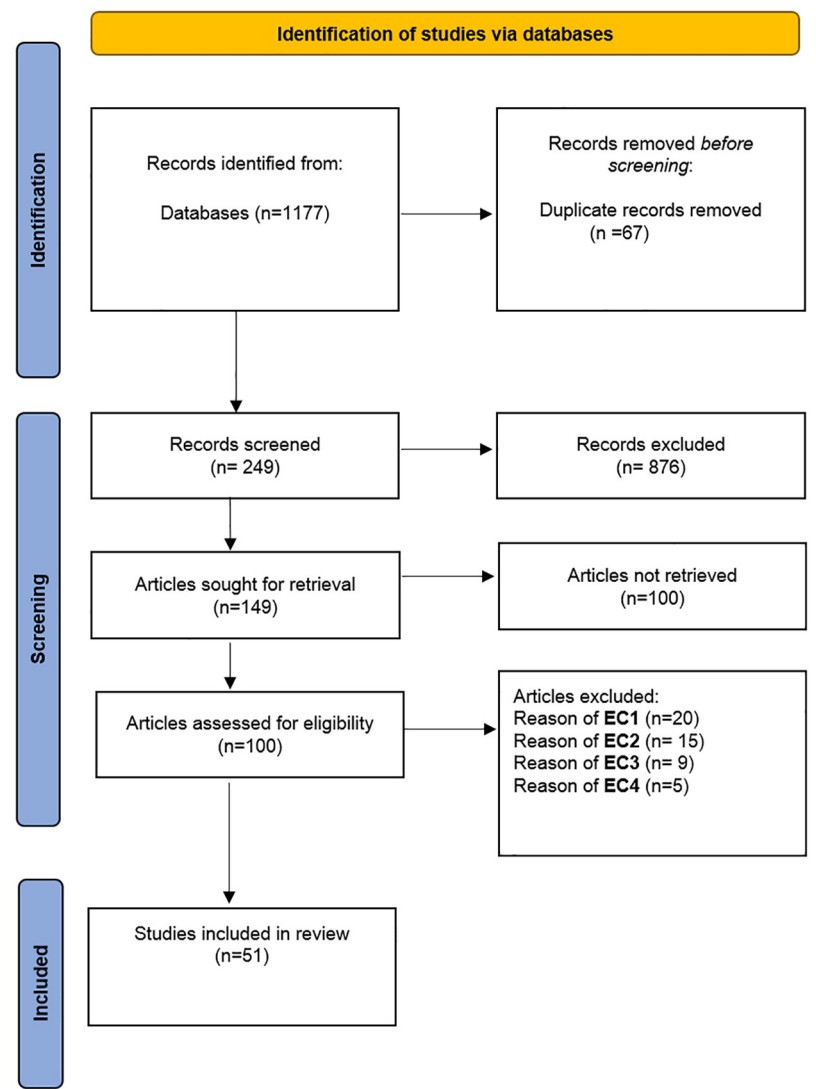

**Fig 1. PRISMA flow chart-based studies selection process.**

were published in 2019, 9 articles were published in 2020 and 2 articles were published in 2021.

The authors of the reviewed articles were found to be affiliated with institutes located across 17 countries. Five countries, China (number of articles = 12), USA (number of articles = 6), South Korea (number of articles = 4), Brazil (number of articles = 3) and India (number of articles = 3), cumulatively represented 65% of the sample (see Fig 3).

In addition, the analysis of author indexed keywords conducted by using word cloud showed that the main emphasis of article related to "blockchain", "technology", "data", "healthcare", "sharing", and "medical" which are graphically illustrated in Fig 4.

**RQ2: What are the major healthcare domains where blockchain technology has been implemented?.** In this part, we will discuss a review of the fundamental principles of blockchain. BCT is used in medicine, especially in managing the information in healthcare that particularly is important in the healthcare area as this technology involves the sensitive data of

**Table 1. List of the selected papers with details of QE, publication channel, year, H-index, and citation per year.**

| Ref | H-index | Publication Channel | Citation per year | | | | |
|---|---|---|---|---|---|---|---|
| | | | 2018 | 2019 | 2020 | 2021 | Total |
| [43] | 68 | Neural Computing and Applications | - | - | 1 | 1 | 2 |
| [44] | 29 | Journal of healthcare and engineering | - | - | - | 1 | 1 |
| [45] | 59 | Procedia Computer Science | - | 10 | 22 | 4 | 36 |
| [46] | 37 | Computational and structural biotechnology journal | 6 | 29 | 26 | 4 | 80 |
| [47] | 52 | Journal of Intelligent & Fuzzy Systems | - | 38 | 21 | 2 | 61 |
| [48] | 43 | Sustainable cities and society | 19 | 67 | 112 | 15 | 242 |
| [49] | 76 | Business Process Management Journal | - | 1 | 7 | 2 | 10 |
| [50] | 153 | Sensors | - | 52 | 119 | 21 | 211 |
| [51] | 67 | IEEE Internet of Things Journal | - | 10 | 13 | 1 | 26 |
| [52] | 70 | Journal of medical systems | 4 | 14 | 10 | - | 30 |
| [53] | 70 | Journal of medical systems | 10 | 69 | 135 | 13 | 245 |
| [54] | 86 | IEEE access | 21 | 74 | 85 | 14 | 227 |
| [55] | 44 | Cognitive Systems Research | 15 | 28 | 42 | 6 | 89 |
| [56] | 20 | Future Internet | - | 2 | 6 | 2 | 12 |
| [57] | 105 | Future Generation Computer Systems | - | 3 | 18 | 5 | 31 |
| [58] | 16 | Electronics | - | 3 | 38 | 11 | 57 |
| [59] | 70 | Journal of medical systems | 3 | 23 | 56 | 1 | 92 |
| [60] | 93 | International Journal of Distributed Sensor Networks | 1 | 7 | 11 | 1 | 10 |
| [61] | 17 | applied sciences | - | 1 | 8 | 1 | 10 |
| [62] | 70 | Journal of medical systems | 4 | 35 | 42 | 9 | 103 |
| [63] | 86 | IEEE Access | - | 5 | 18 | 3 | 28 |
| [64] | 108 | Oncotarget | 18 | 56 | 57 | 11 | 154 |
| [65] | 68 | The Journal of Behavioral Health Services & Research | 1 | 2 | 28 | 2 | 37 |
| [66] | 86 | IEEE access | - | 9 | 49 | 4 | 74 |
| [67] | 105 | Future generation computer systems | 1 | 17 | 56 | 8 | 88 |
| [68] | 60 | Wireless Communications and Mobile Computing | - | 1 | 4 | 1 | 8 |
| [69] | 17 | Applied sciences | 1 | 6 | 39 | 11 | 62 |
| [70] | 70 | Journal of medical systems | - | 11 | 14 | 3 | 32 |
| [71] | 86 | IEEE Access | 1 | 23 | 39 | 14 | 83 |
| [72] | 70 | Applied Innovation | 2 | 34 | 46 | 8 | 103 |
| [73] | 298 | Nature communications | 1 | 11 | 32 | 4 | 49 |
| [74] | 17 | Applied Sciences | - | 7 | 14 | 2 | 22 |
| [75] | 70 | Journal of medical systems | 5 | 41 | 59 | 12 | 136 |
| [28] | 37 | Computational and structural biotechnology journal | 10 | 78 | 120 | 9 | 237 |
| [76] | 127 | Journal of medical Internet research | - | 3 | 25 | 2 | 31 |
| [77] | 70 | Journal of medical systems | 5 | 21 | 26 | 4 | 57 |
| [15] | 99 | International Journal of Medical Informatics | - | - | 29 | 9 | 46 |
| [78] | 93 | Computers in Industry | - | - | 7 | 6 | 13 |
| [79] | 86 | IEEE Access | - | - | 12 | 4 | 16 |
| [80] | 153 | Sensors | - | - | 13 | 3 | 17 |
| [31] | 89 | IEEE Transactions on Engineering Management. | - | - | 4 | 2 | 6 |
| [81] | N/A | International Journal of Environmental Research and Public Health | - | - | - | 4 | 4 |
| [82] | N/A | International Journal of Healthcare Information Systems and Informatics | - | - | 1 | 2 | 3 |
| [83] | 86 | IEEE Access | - | - | 6 | 3 | 10 |
| [84] | - | Electronics | - | - | 27 | 8 | 38 |
| [71] | 86 | IEEE Access | 1 | 23 | 42 | 12 | 83 |

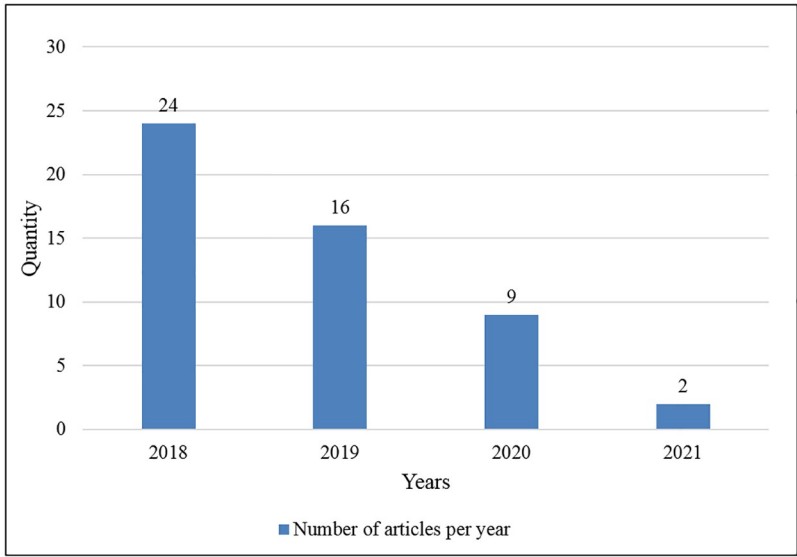

**Fig 2. Number of articles published per year.**

patients. This sector is important to society because innovations in this field will enhance the quality of people's life. Following this logic, the computation can help to mitigate the effects of certain problems in this field. Informatics, for example, helps in the automation of medical records by ensuring more reliable data sharing, log management, and other applications. One of the first and most popular blockchain applications in healthcare is the exchange of health records. Information related to health is difficult to disclose because it is labeled as confidential information and includes patients' details. Among the key works in the literature that discuss this application of blockchain technology are: [85, 86]. The characteristics of blockchain-based architectures for the sharing of electronic healthcare records can vary. The features of

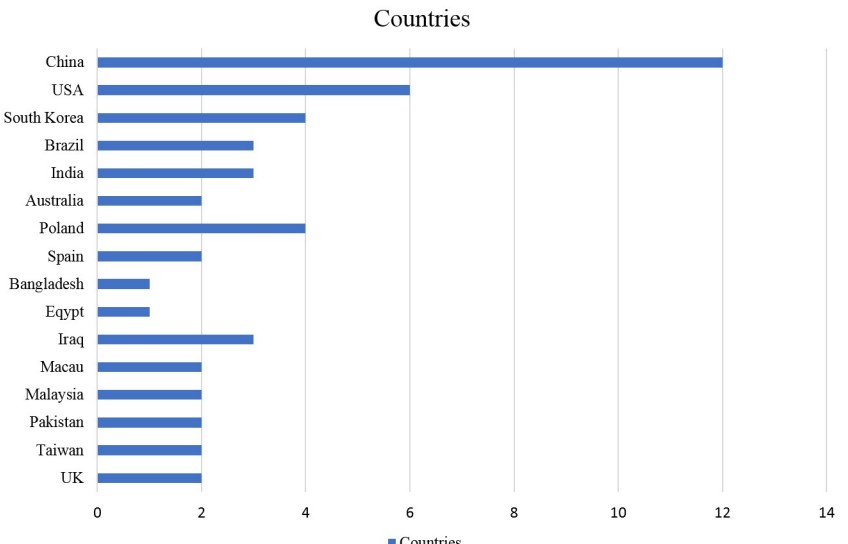

**Fig 3. Country-wise publications of selected research studies.**

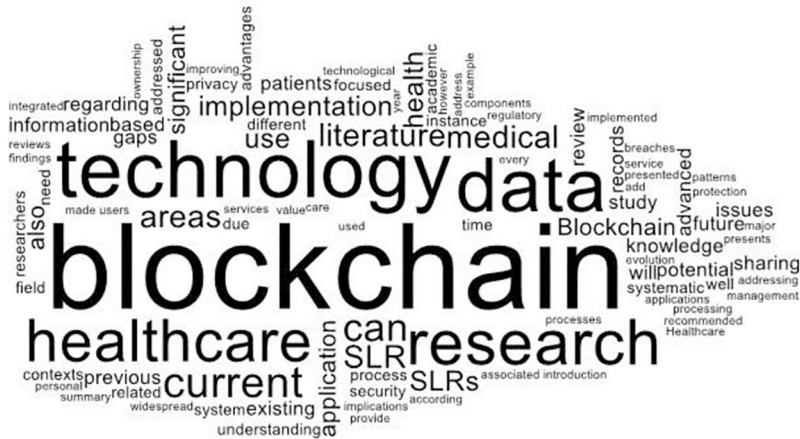

**Fig 4. Constructs key framework.**

blockchain-based systems for the exchange of electronic healthcare records may vary. One of the most well-known structures in the literature is discussed in the work of Azaria et al. [86]. Several recent papers in the literature have cited this as a framework for the development of other similar architectures. Some of these systems are inspired by Azaria et al. [86] cited in [30, 79, 87, 88]. Voting is a formal statement of an individual's or a group's opinion or choice, whether positive or negative. Traditional voting methods, on the other hand, are centralized and are known to have security and efficiency flaws. The study [89] examines blockchain-based voting systems in depth and categorizes them based on a variety of characteristics e.g., the types of blockchain used, the consensus approaches used, and the scale of participants. Artificial intelligence (AI) is now the core technology for a wide range of applications, from self-driving cars to smart cities. One of the most crucial pillars of social and economic stability is smart healthcare, which is an integral part of smart cities. The research study [90] focused on designing a human-in-the-loop-aided (HitL-aided) scheme to protect patient privacy in smart healthcare. Profile matching technology can facilitate the sharing of medical information across patients by matching similar symptom traits. However, because the symptom attributes are linked to sensitive information about patients, their privacy will be compromised during the IoMT matching process. To accomplish fine-grained profile matching, the study [91] provides a verifiable private set intersection scheme and used a re-encryption technique to preserve patients' privacy. Technologically advanced countries are exploring or implementing smart homes, it is convenient but risky. Most of the existing solutions are generally based on a single-server architecture, which has limitations in terms of privacy, integrity, and confidentiality. While blockchain-based solutions may alleviate some of these problems, they still face some significant obstacles. Lin et al. [92] developed a revolutionary safe mutual authentication method for use in smart homes and other applications. MedRec will be the first sharing architecture to be discussed, which uses a blockchain-based system to store electronic medical records. The MedRec considers resolving issues as data access response time, interoperability, and increased data quality in healthcare research [86]. It is worth looking into the resources that were used to create MedRec's architecture, since it implements a private P2P network (Permission block chain), as well as using Ethereum's smart contract platform, to make it easier to monitor and track network state transitions. One of the MedRec architecture's hallmarks is that it provides patients with a consulting agency that has records of their healthcare

background, enabling them to remain informed about health decisions. Another difference is that they enable the standardization of health data since they are adaptable and provide open data standards in a variety of formats. This architecture takes a novel approach to the use of health data management systems by enhancing security and establishing a common language for data exchange for research purposes [86]. While Azaria et al. [86] also plan to perform experiments and analyses with a diverse community of users. In summary, MedRec is a realistic choice for exchanging healthcare information that can be used to combine patient care, hospital care, and physician care. As a consequence, the reported data can help to minimize discrepancies among different systems of hospitals. As stated by [85], the method introduces the topic of cloud computing, which could help in creating new architectures for sharing healthcare records via blockchain, resulting in safer and more secure healthcare systems for clinical use. The authors propose a cloud-based architecture that uses a blockchain-based data system to connect a network of communication nodes. The paper [85] shows how to handle the exchange of healthcare information using a blockchain architecture, which employs the principles of intelligent contracts and, immutable bookkeeping. The major roles of BCT in sharing health information, remote care with IoT, security, and privacy, and supply chain are depicted in Fig 5.

The list of blockchain-based healthcare methods is discussed in Table 2.

**RQ3: What are the existing problems and constraints raised by the previous studies in the healthcare field using blockchain technology?.** Even though blockchain is a multidisciplinary concept with challenges and limitations, it can be applied to a variety of areas [88]. Researchers in this field are working to overcome or mitigate the negative effects of these factors. The following are some of the problems (i.e. technological challenges) that blockchain technology faces when used in healthcare [22, 88, 97, 98].

**1) Throughput.** If the number of transactions and nodes in the network grows, more checks will need to be performed, possibly causing a network bottleneck. When dealing with healthcare systems, high throughput is a challenge because unless there is fast access, it might adversely affect a diagnosis which could save someone's life [64], correspondingly recognizing that the suggested framework focuses on identifying inconsistencies will possibly not perform well when datasets are unlabeled. Issues including specifications for continuous updates by the used system [59], keyword set size [75], network set-up and disk space needed based on the blockchain type, such as Ethereum software, employed to the framework can all affect a framework's scalability and performance quality [25, 99]. Similarly [48], suggests that integrating certain features into established systems, e.g. making global smart deals, can offset higher performance-related costs. Furthermore, a small number of studies have suggested that performance-related problems can be connected to the node management in a suggested system.

**2) Latency.** Validating a block takes about 10 minutes; this can be harmful to system security services since successful attacks may occur during that time. Healthcare networks are complex and should be accessed at all times, as any delay may negatively affect the analysis of an exam.

**3) Security.** When a party has control of 51 percent of the voting power, this can adversely affect the computing power of the network. This is a serious issue that needs to be addressed because a harmed healthcare system will lead to healthcare organizations losing their reputation.

**4) Resource Consumption.** Since the mining process consumes a lot of energy, using this technology could result in a significant loss of resources. Since multiple devices are required to track patients in a healthcare setting, energy costs are high; however, the use of blockchain may result in high computing and energy costs. Managing these expenses is a challenge for businesses.

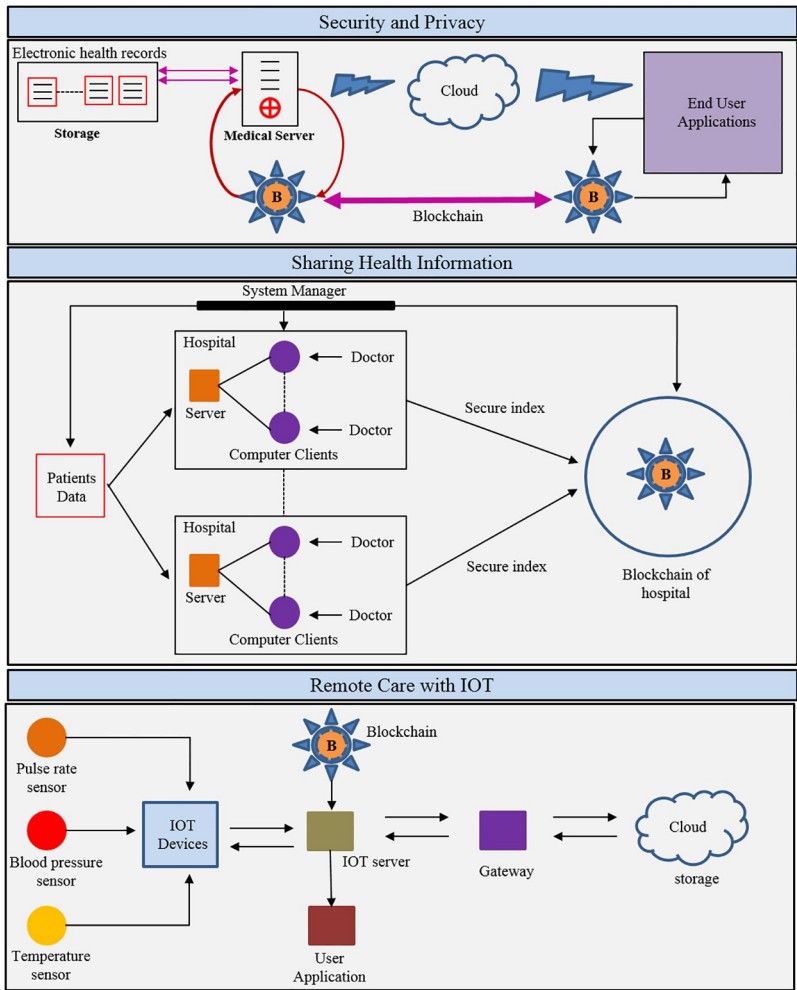

**Fig 5. Major healthcare domains where BCT has been used.**

**5) Usability.** Since these systems are so complicated, usability is a challenge as well to deal with. Additionally, an API must be developed (Application Programming Interface) Users would enjoy the user-friendly features. Since not all health practitioners have the same level of education, As IT professionals, we should be able to use frameworks that are easy and effective.

**Table 2. Blockchain-based healthcare methods.**

| Ref | Domain | Methods |
|-----|--------|---------|
| [86] | Sharing Health Information | MedRec |
| [93] | Sharing Health Information | MedRec |
| [94] | Sharing Health Information | Medicalchain |
| [71] | Remote Care with IoT | Patient-Centric agent (PCA) |
| [95] | Supply Chain for healthcare | Modum |
| [96] | Security and Privacy | Decentralized sharing of health records (DSHR) |

**6) Centralization.** Even though blockchain has a decentralized design, certain implementations tend to concentrate the miner, which decreases network stability. Because this central node is insecure and may be hacked, hostile attackers can get access to the data it holds [25].

**7) Privacy.** It is common to suppose that the Bitcoin framework allows blockchain to make sure the privacy of its nodes. The results of [25], on the other hand, contradict this assumption. Furthermore, strategies to provide this functionality to blockchain-based systems are needed [25]. Due to privacy laws and regulations, blockchain-based systems have to conform to the General Data Protection Regulation (GDPR). Our research also indicates those users' reservations about the safe and ethical utilization of data could be a major barrier to blockchain adoption in healthcare systems. The existing issues are primarily associated with blockchain technology's technological limitations, such as the protection of individual nodes [53], the degree of safety permitted through cryptographic elements implemented with the system [70], besides the preservation of confidential data whereas requesters complete their computations [100]. However, certain research has drawn attention to more socially relevant issues regarding sharing of public data [73] and users' confidence in governments [49, 74]. Such issues may also be linked to the suggested framework protection from the perspective of users for example users' management and misuse of permitted personal keys/codes [46].

**8) Constraint.** Prior studies have identified constraints, which can be divided into four categories. These dimensions mean that such constraints extend beyond technical boundaries (costs of designing, implementing blockchain systems, data analysis for system assessment, and framework constituent elements) to include certain social facets also such as trust in the administration, infrastructure of technology in a country.

**9) Costs.** This set of constraints is specifically concerned with the time, capital, and economic expenses of putting a system of blockchain into action. For example [50], discuss resource constraints in IoT, while [28] discuss the costs of arranging dispersed app in the deployment of blockchain. Additional expenses that have been established as constraints and limitations in previous research include the linear increase in protocol costs based on the characteristics and attributes of the entities involved, such as patients [54], increased operational overhead for the patient, and access latency for the requester [69], the exchange and implementation costs depend upon inconstant inputs in size and length of a string [67]. The issues related to time are further listed as one of the limitations, i.e. the spent time in finding smart contracts globally [48], increased time consumption [57], transmission timing [53], the time needed for the data receiver to seek the required data in shared storage [68], and higher overall execution time [21].

**RQ4: What are the potential healthcare avenues that would benefit from blockchain technology implementation?.** Blockchain consists of a sequence of blocks connected with cryptographic techniques. The immutability of this is one of the most attractive characteristics to many industries. The data that is added to the blockchain is irreversible, consequently, allowing for the creation of a consensus-based, verifiable, and accurate data ledger. That creates blockchain especially well-suited to tasks wherever integrity of data is critical; ProvChain [101], an infrastructure based on this technology in giving chain-of-custody to the database, is a functional example of this immutability. There are many blockchain implementations, including Bitcoin, a cryptocurrency token based on the blockchain; and Ethereum, a cryptocurrency token based on blockchain. Ethereum [102], a blockchain ledger with Turing-complete computer-generated device that allows smart contracts to implement code on this; and JP Morgan's Juno [103], an Ethereum fork that uses the particular consensus mechanism called Quorum, along with several other blockchain implementations. The execution of blockchain varies due to ways in their consensus approaches. Bitcoin, for example, employs the HashCash [104] Proof-Of-Work algorithm, which is a deliberately slow system intended to

avoid denial-of-service attacks. As a vote against the blockchain's agreement, every Bitcoin miner authenticates this blockchain system by conducting that algorithm. Ethereum includes Ethash, which is an algorithm called Proof-of-work based on the Dagger-Hashimoto algorithm, as described in the Ethereum Yellow Paper [103]. However, shortly Ethereum is likely to advance in an algorithm named Casper. It will consider the excess requirement of energy in Proof-of-work [105]. The implementation of smart contracts separates Ethereum from Bitcoin. The smart contract is one of the snippets of code that run on each blockchain node. These are self-executing contracts in which all members of the blockchain are bound by the agreement. In the same way, as a standard contract does, they influence advantages, responsibilities, also punishments related to contract-related conduct. It could be utilized to model the HIPAA healthcare personal health information (PHI) workflow to satisfy audit and regulatory standards, likewise, done inside Patientory since they resemble conventional paper contracts and rules [106]. A new type of blockchain trust model, trust in the consortium, is also emerging. Microsoft recently released the Coco framework, which enables the creation of blockchain-agnostic consortiums [107]. Above mentioned models are based on a pre-defined group of trustworthy parties. It can be among various clinics or in the UK, NHS Trusts, third parties, and manufacturers of devices. By implementing smart contracts only on the hardware of trusted partners, without requiring miners, a consensus can be generated. It turned out in remarkably improved results, through a Coco-optimized blockchain case capable of processing 1600 transactions in a second, taking the blockchain system very close to the major payment processors. Coco also supports a variety of trusted execution environments, including Windows Virtual Secure Mode, Arm Trust Zone, and Intel Guard Extensions to name a few.

**1) Clinical trials.** Managing trial subject consent and clinical trials itself is an area in which blockchain can potentially improve the accountability, audit ability, and transparency of researchers and practitioners in the medical field. By keeping the unchangeable log of a patient's approval, officials could control the standard of clinical trials easily, making sure it complies with informed consent regulations of the country. It is especially important because a forged informed consent form is one of the common types of clinical fraud. It involves falsifying patient consent and editing records, implying that authentication of trial subjects is essential for avoiding it. That kind of setup may be improved by implementing a smart contract system that stops clinicians to use the data of patients unless a key is issued by the end of an auditable process of smart contract that requires permission in each step in the trial, as proposed by Benchoufi, Porcher, and Ravaud. This procedure should also allow the patient's consent to be revoked. Executing the clinical trial of blockchain consent log provides the subjects with data ownership while also having a trail of audit for regulators, medical professionals, and researchers. The role of BCT in clinical trials is graphically represented in Fig 6.

**2) Sharing the data.** Sharing information is regarded as the most significant opportunity for improvement in healthcare; however, it too poses a significant challenge in privacy. Sharing the data using BCT is presented in Fig 7. Powles and Hodson [108] use DeepMind's case study teamwork with the Royal Free London NHS Foundation Trust to address the need for transparency in how patient data is being shared with third parties. Regardless of the good impact on diagnosis/treatment of patients by the product suite of Google, one of the significant issues addressed in the previous case study was a lack of patient consent. On the other hand, Sleep Apnea American Association, and IBM [109] were collaborating to solve major healthcare challenges to examine sleep apnea (with IBM's Watson supercomputer at home) in thousands of Americans, with informed and clear patient consent. That was critical to implementing the national standard for interoperability in the healthcare system of IT. Which was emphasized by Wachter and Hafter through a white paper in UK NHS in comparison to the US healthcare sector that emphasized the significance of interoperability in permitting patient Electronic

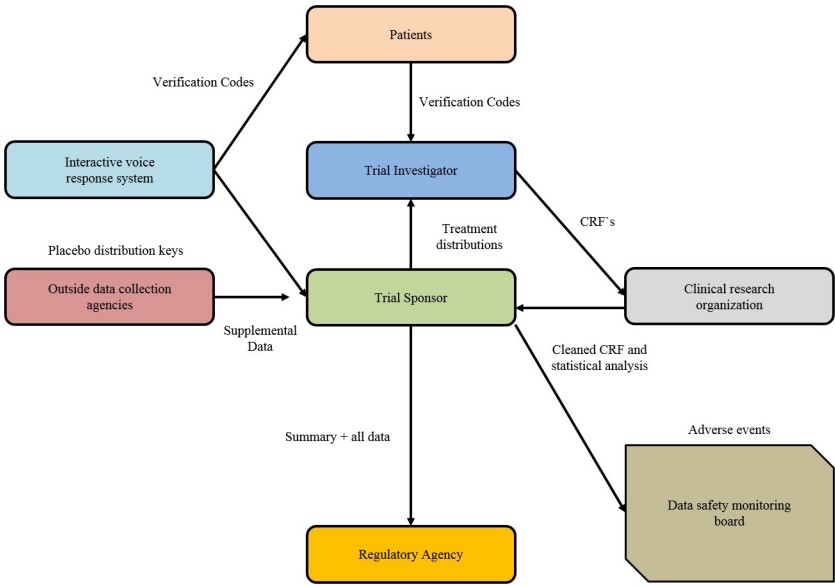

**Fig 6. Role of blockchain in clinical trials.**

Health Records (EHRs) over various clinics, such as various trusts that do not maintain a separate system to get access on these records built by different vendors.

In a report of Harland Simon on a project justifying RFID tagging in NHS Cambridge shire, about 15% loss of assets annually, resulting in a substantial cost to repurchase the items that hospitals previously have. Furthermore, as per GE Healthcare [110] report, nurses spend an average of 21 minutes per shift searching for misplaced devices as stated by, defining any device under $5000 as consumable and to again purchase if any device is lost, suggesting

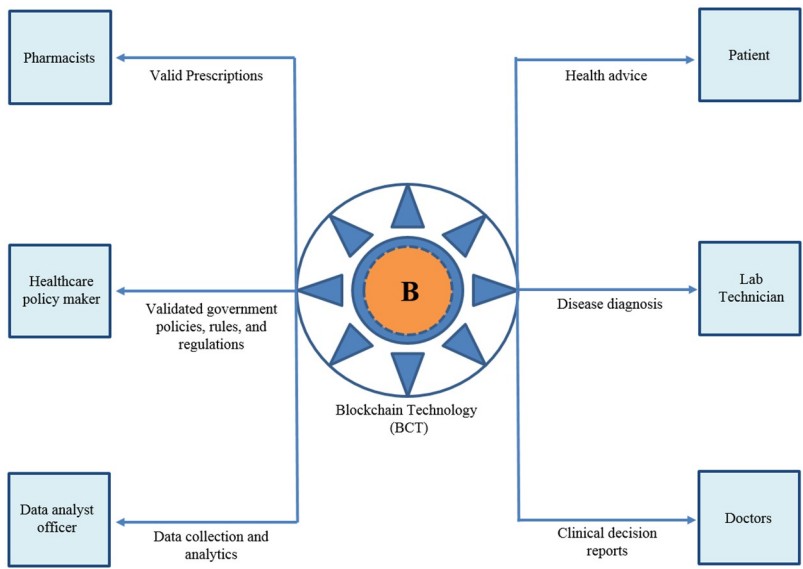

**Fig 7. Sharing health information using BCT.**

significant cost in the sector. Published by Harland Simon, another study reported that [111], by adopting radio frequency identification (RFID) standards for tracking of medical devices, NHS Forth Valley in Scotland had saved nearly £400,000 in cost avoidance by not having to buy the important devices which would have been lost by the medical system. Tracking of Drugs has been a completely different issue than tracking of devices because a major concern is counterfeits of drugs here. According to WHO's report, In the US up to 10% supply of Pharmaceutical products is counterfeit. In the United States, the Food and Drug Administration (FDA) recently approved the utilizing the RFID to track pharmaceuticals between the supply side to the patient. It enables the whole sequence kept supervised, to make sure that pharmaceuticals were purchased from a legitimate source. Pfizer was the first pharmaceutical company to use RFID "e-pedigree" to ensure that patients and doctors could trust the source and capabilities of their flagship medicine, Viagra, after identifying it as one of their most counterfeited drugs. Because of the use of low-cost passive RFID tags and barcodes, the system enabled the pharmacists and wholesalers to check the authenticity of their Viagra through a simple RFID scanner at a low rate than Pfizer.

**3) Records of patients.** Blockchain is having the potential to significantly disrupt health services and place data in the patient's hand. The specific intriguing steps are in MedRec [86], which provides doctors and patients with an immutable log of a health record as shown in Fig 8. That has a different approach to incentivize miners by providing access to anonymized data about health in exchange for network maintenance. MedRec maps Patient-Provider Relationships (PPRs) using Smart Contracts when the contract displays a reference list having relationships between nodes on the Blockchain-system. This too places PPRs in the patient's hand, empowering them in accepting, rejecting, or modifying relations with health service providers for example doctors, insurers, and hospitals, etc. Blockchain-system allows for interoperability in the health system by providing a decentralized ledger of accepted facts in healthcare records to which all health service providers are having access. It implies that while user interfaces may differ, the central ledger would be the same across all service suppliers. A challenge that exists

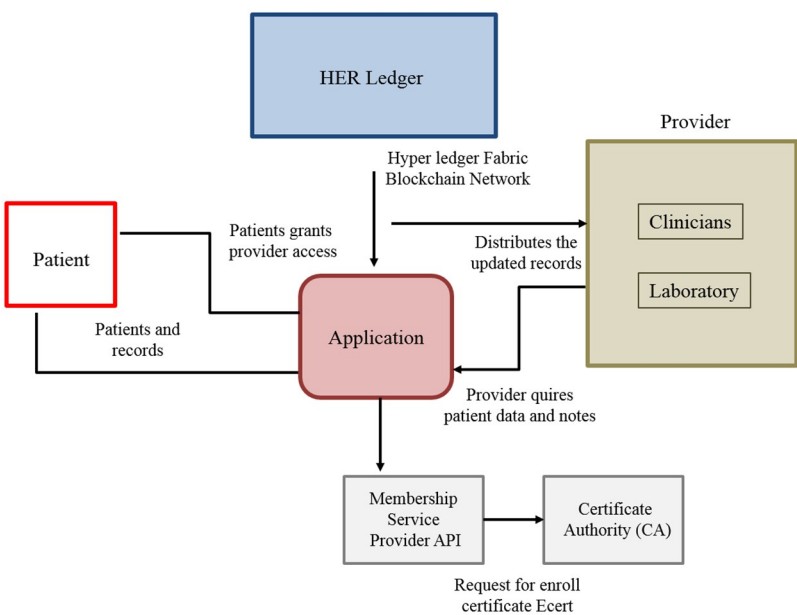

**Fig 8. Records of patients.**

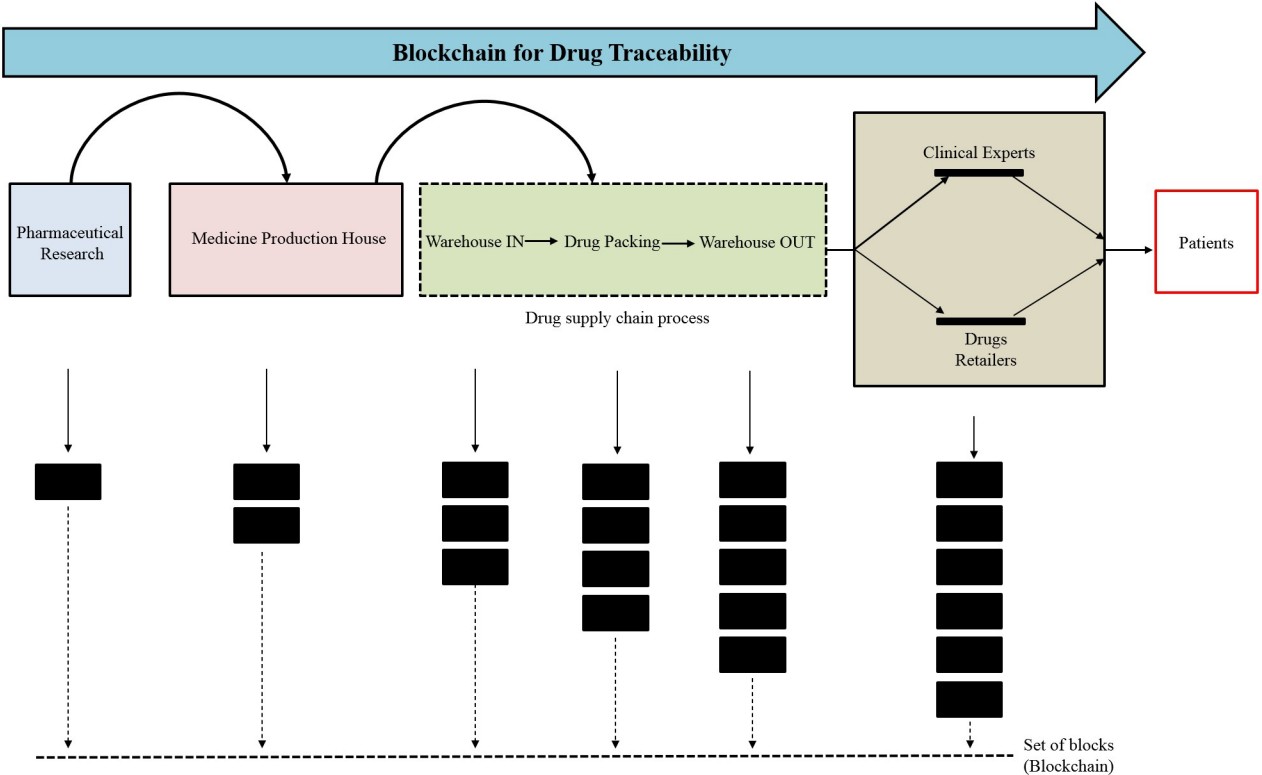

**Fig 9. Drug tracking.**

relates to the current state of health records across providers, which contain significant amounts of the same information under different identifiers that may not be linked. This causes replication, and as the blockchain system increases in size, it is reduced in performance. The level of data duplication in all records will necessitate replication to maintain a reasonably performant system with unique, anonymized identifiers to identify the patient in all kinds of service. Adopting the blockchain health record is a business challenge in and of itself. The important thing is that medical records will not start from zero because they would have to replace the current setup, and that is challenging. Furthermore, the sheer volume of data generated in the healthcare sector is ever-growing, with Kaiser Permanente estimated to have between 26 and 44 petabytes of data on its 9 million members from EHRs and other medical data in 2014. The data volume which is logged and referenced would mainly exacerbate the scalability issue.

**4) Drug tracking.** Another opportunity is tracking the drugs using a blockchain system as shown in Fig 9, which takes advantage of its immutability in the development of tracking and a chain of custody from manufacturer to patient. Chronicled is a technology startup company that is working on its product, Discover, that develops a chain of custody model that shows the manufacturing place of a drug, the places it had been since then, and when it was disbursed to patients, hence reducing the pharmaceutical theft and fraud. That enables the health professionals in meeting existing standards of the pharmaceutical supply chain, along with focusing on interoperability among healthcare professionals. The Counterfeit Medicines Project has been launched by Hyperledger, the Open-Source Blockchain Working Group, to address the issue of counterfeits of medicines. The origins of counterfeit medicines would have been

tracked and thus eliminated from the chain of supply. One benefit of tracking drugs by block-chain-system over conventional methods is the inherent decentralization of trust and authority in the technology's principles; whereas chief authorities could have bribed or faked, it is much more difficult to bribe a consensus of those on the blockchain. As a result, an existing standard in pharmaceuticals tracking in industry, ePedigree, which already employs RFID and a traditional database, is transitioning to its blockchain application. If medicines/drugs could be tracked and developed at the point of manufacture using blockchain's inherent anti-tampering capabilities, that will remove the counterfeited pharma products engaging in the supply chain.

**5) Device tracking.** Tracking of medical devices is one aspect of Block chain in disrupting healthcare, from manufacturing to decommissioning. The monetary benefit generated through tracking of assets is clear; NHS East Kent Hospital discovered 98 infusion pumps they had no idea they still owned across three sites as concluded through the case study of Harland Simon [111] in which active RFID trackers were implanted. Because of this single case study, they saved $147,000 at $1,500 per person. The use of blockchain in conjunction with this technology allows for an immutable ledger that not only shows the current location of the device, also the location of the lifecycle, along with the serial number, distributors, and the manufacturer linked with the device, assisting with regulatory compliance. Deloitte identified the competency among the potential game-changers for blockchain in the domain of healthcare in a white paper. According to an IBM study, 60 percent of government stakeholders in healthcare believed the integration of medical devices and asset management as the most likely area of disturbance in industry. The blockchain-based system has various advantages above conventional products of location tracking. This immutability and tamper-proof properties of the Blockchain are the most obvious. This prevents a malicious user from changing or deleting a device's location history. This is especially important given that theft of devices and shrinkage has been a major issue in the United States and the United Kingdom. This immutability, in addition to preventing traditional theft, also protect devices from being lost and reordered, that have incurred high cost in terms of both actual equipment cost and care provided. The setup must not add significantly to the workload of staff, nurses, or workers because that requires tapping the device only using a mobile phone and further entering the device location. Whereas the use of blockchain on the Internet of Things (IoT) is still in its early stages, Huh states a method to communicate the devices using an Ethereum blockchain and public key system of RSA. Likewise, the device while it stores on blockchain its public key also stores the associated private key on the device.

## Discussion

This study aimed to conduct an organized analysis of previous literature about the employment blockchain in healthcare for a better understanding of their current and probable state. The four key research problems are defined for this reason. RQ1 was presented with summing up top writers, publishers, publication houses, and designs of publication patterns of this subject. Furthermore, it included an existing outline of research about employing blockchain in the healthcare space. The comprehensive description of the reviewed articles is discussed in Table 3. RQ2 was designed to help researchers better understand how blockchain can be used, it is responded by defining particular themes and sub-themes that reflect key aspects in employing blockchain in this sector. RQ3 further discussed its shortcomings and obstacles that previous researchers had encountered. We were able to recognize the research gap in the existing literature and responded to this question by summarizing its main research themes and existing limitations. RQ4 concentrated on the key aspects where future investigation can

**Table 3. Comprehensive description of reviewed studies.**

| Ref | Methods | Description | Channel | Proposed / Implemented | Components |
|---|---|---|---|---|---|
| [45] | Public blockchain | Details about the algorithm are provided. | multi-tier, public | Implemented | PBEDA, ECDH, MVP, and ECDSA |
| [46] | MAM is used for real-time broadcasting activity via wearable's | Algorithm details are provided. | IOTA | Implemented | Masked Authenticated Messaging for Tangle, Merkle hash technique, Merkle signature scheme, One-time signature |
| [47] | Computing Edge | There was little information available on the outcomes and efficiency. | The channel was Ethereum | Proposed | WSN and Wireless sensor network controller v.2, computing Edge |
| [48] | Not Available | MedRec was used to do a cost analysis, but no algorithm or data were given. | The channel was Ethereum | Proposed | Ethereum Go-client, QuorumChain algorithm, classification, smart contracts, cipher manager, proxy re-encryption |
| [49] | UID system | Details about the algorithm are not provided. | Ethereum channel | Proposed | UID system, analytics of Big Data, history & registration contract |
| [50] | Not Available | Details of the algorithm are provided, along with security margins that have been verified against defined criteria. | The channel was Bitcoin | Proposed | Internet of Things, Merkle tree, Diffie–Hellman key exchange, and digital ring signature |
| [51] | Not Available | Details about the algorithm are provided. | The channel was Consortium | Implemented | Practical Byzantine fault tolerance consensus mechanism, |
| [52] | PSO | Performance results from a static study based on specific assessment criteria | N/A | Implemented | ADB, boosting for ML, reverse engineering, feature selection & extraction |
| [53] | Not Available | Details about the algorithm are not provided. no experiments. | Federated blockchain | Proposed | IoT sensors WBANs, oracle |
| [54] | Proposed signature scheme based on Attribute | Details of mathematical and computational notations given for execution of scheme, performance, and security evaluation | Not Available | Implemented | Diffie-Hellman computational bilinear, MA-ABS scheme |
| [55] | With the support of a genetic algorithm Proposed access control method based on blockchain | In terms of input and output strings, block creation, time processing, detailed simulation results are provided. | N/A | Implemented | Genetic algorithm, hash key cryptography, MD5 strings, discrete wavelet transforms. |
| [56] | The proposed Blockchain-based eHealth Integrity Model uses a design-science methodology. | The integrity-verification algorithm's details and test results have been released. | Permissioned blockchain | Implemented | Byzantine Fault in Practice Algorithm for tolerant consensus |
| [57] | (HAR) Methods for recognizing human activity based on a uni model | Some mathematical notations and comprehensive findings for performance reassessment studies on three datasets have been given. | N/A | Implemented | For HAR (ECOC) framework, (SVM), Multi-class cooperative categorization technique, fog computing |
| [33] | Not Available | For performance evaluation, precise mathematical and algorithmic notations are supplied, as well as the outcomes of experiments. | N/A | Implemented | The Merkle tree with order-preserving encryption |
| [59] | Not Available | The applicability of blockchain in healthcare is discussed using a concept-based approach. | N/A | Proposed | Environment based on the blockchain |
| [60] | Preprocessing of images | Details on the experimental results, classification training, and testing outcomes were given. | Bitcoin | Implemented | HOG, LBP, SVM, RFT, DNN |
| [61] | Pearson's correlation, compression ratio technique | The compression ratio and stability performance testing results were given. | N/A | Implemented | (BAQALC) proposed, (NGS), (SRA), LZW modification |
| [62] | Methods of authentication | Details on the algorithm and how it performs in areas of operational costs were given. | Ethereum | Implemented | primitiveness verification (PV), preservation Submission |
| [63] | Method of the primal-dual Varangian | The performance study results of a strategy for achieving Stackelberg equilibrium were disclosed. | N/A | Implemented | three-layer (Hierarchical architecture), edge computing |
| [64] | Machine Learning(ML) | In place of simulation, experiment-based assessment, the proposed architecture is presented through workflow examples. | Exonum | Proposed | Inbreeding coefficient, DNN predictor, data temporal value, LifePound (utility crypto token) |

*(Continued)*

**Table 3.** (Continued)

| Ref | Methods | Description | Channel | Proposed / Implemented | Components |
|---|---|---|---|---|---|
| [65] | Not Available | There are just a few algorithmic notations, but comprehensive findings for Apache JMeter performance evaluation are supplied. | N/A | Implemented | Merkle tree, timestamped algorithm, Keyless signature infrastructure |
| [66] | Not Available | Code scripts, access control methods, and performance assessment findings for access control and network overheads are all shared. | Permissioned blockchain on Ethereum | Implemented | (IPFS), (ABE), mobile cloud |
| [67] | Not Available | For performance evaluation on many parameters, details of algorithmic notations are given together with findings. | Permissioned blockchain on Ethereum | Implemented | MediBChain protocol, (ECC) |
| [68] | Not Available | For performance parameters, there are just a few algorithmic notations, but extensive simulation results are supplied | N/A | Implemented | The one-time transaction, Ring signature algorithm, stealth address, Cryptonote protocol |
| [69] | Not Available | For numerous parameters, with the discussion of outcomes, there are few algorithmic notations from experiment-based and theoretical mathematical results | Permissioned blockchain | Implemented | consensus algorithm BFT smart, ECC, MedChain, modified digest algorithm |
| [70] | Not Available | For simulated security analysis and performance evaluation, few explained algorithmic notation results are used. | Blockchain technology Hyper ledger | Implemented | SIFF |
| [71] | Generating Sessional symmetric key | An in-depth look into simulated performance and security assessments. | Custom bitcoin and Ethereum | Implemented | PUA, Trei tree, mutual authentication protocol, HMAC |
| [72] | Not Available | There are no experimental or simulation test results, but algorithmic notations and the syntax are described with analysis of specified security. | Consortium blockchain | Implemented | ABE and IBE, proposed d identity-based combine attribute, identity-based signature, signature, and encryption |
| [73] | Not Available | There was only a brief discussion of the outcomes of simulated blockchain-based clinical trials. There is no explanation for the algorithm or syntax. | N/A | Implemented | artificial healthcare, Parallel healthcare system, IVRS, parallel execution |
| [74] | Not Available | There is only a brief explanation of the method and the findings of the prototype implementation, which is mostly theoretical. | Consortium blockchain technology | Proposed | proposed proof of familiarity(PoF), API |
| [75] | Proposed protocol privacy-preserving and Secure PHI sharing BSPP | Algorithm notations, system architecture, and protocol implementation, and performance assessment findings are all detailed. | Private, Consortium | Implemented | Bilinear maps, Consensus mechanism |
| [28] | Case study involves | Through a case study, a detailed explanation of the proposed architecture and process is provided. There are no experiments or discussions on the algorithm or syntax. | Ethereum technology | Proposed | Oauth, Public-key cryptography—sign then encrypt mechanism, Solidity smart contract, FHIR, |
| [76] | Not Available | There is not much explanation of the algorithm, but there are a lot of details about the experiment that was done to verify the system's feasibility. | IOTA Tangle (DLT) | Implemented | (MAM) Masked authenticated messaging, GPS, IoT integration |
| [77] | Not Available | Algorithm creation and performance assessment for processing time and transaction verification are discussed. | Ethereum technology | Implemented | MIStore, PBFT |

provide valuable insight. The fourth research question is addressed by combining findings through emerging differences, shortcomings, and previously proposed guidelines.

## Conceptual evolution

According to the findings of the study, research in healthcare's blockchain was largely focused on enhancing more creating new ideas and concepts that help researchers to derive multi-

domain [59] also practicable blockchain in healthcare implementations. The viability of employment [99] is being established and evaluated across three sub-themes of research: design creation, applications on benefit-based, and developing predictive competencies.

## Concept development

The findings of the analysis indicate that new proofs and algorithms have received a lot of attention, such as proof of data primitiveness [62, 112], proof of familiarity [74], and simpler workload for proof of work; [54]. Studies have also focused on testing new variables and components in architecture systems, as well as improving frameworks that enable blockchain execution by including them. Consider cryptosystems based on attribute [72], approach the Stackelberg game [63], sibling intractable functions [70], and homomorphic computations for more efficient frameworks [77]. Further [113], suggested a new scheme (BBDS) based on blockchain to protect data transactions and maintain privacy [57]. Used fog computing estimation efficiency as well as reliable models for human pursuit acknowledgment to support remote e-health controlling [51]. To eliminate a single point of failure, they are focused on incorporating several time sources into their technique. Finally, this research has centered on how to boost the efficiency of already established algorithms and structures based on the blockchain.

## Benefit-based application

Blockchain has been used in healthcare research to extract concrete benefits by identifying and testing new technology avenues. It involves work in upgrading the technical advantages of employing blockchains, such as advanced image processing [60], effective behavior recognition [57], and Internet-of-Things synchronization (IoT) devices [51]. Furthermore, the majority of studies in this category have centered on the use of blockchain technology to establish specific benefits of healthcare, e.g. mutual decision making in the medical field [74]. Blockchain adoption, for example, is being suggested to have positive implications while managing clinical trials [73], DNA data transmission [61], preventive healthcare, biomarker growth, and discovery of drugs are all examples of remote patient monitoring [50, 53, 64].

## Advancing decentralization

Existing research is also considering promoting key advantages of blockchain technology throughout healthcare environments for encouraging justice and also efficient decentralization [76, 114]. For instance [63], produced an efficient framework for promoting maximization of revenue maximization along with fair decentralized trade, considering that [48] stated the need for trade-offs for mining benefits. Researchers in previous literature already described blockchain's prospects in developing transparency in exchanging the data [56], such as utilizing upright client roles [21, 30]. By doing this, we can say that previous research about the increasing use of blockchain-based technology in the medical space is focused on spreading decentralization along with its related advantages.

## Advancement in technology

Current studies have contributed substantial progress in terms of advancement and refinement of blockchain for the development of targeted deployment, particularly in the healthcare space. We suggested previous research that is classified in this theme be directed regarding the three main topical issues based on our review:

## Developing intelligent healthcare ecosystems

The introduction of blockchain technology programs into healthcare environments has piqued the interest of some academics [45]. Such integrations can pave the way for the development of intelligent healthcare systems [100]. For example [47], argues that blockchain adoption will aid in the development of a more efficient e-health ecosystem. Prior research has also suggested frameworks for developing blockchain-based e-health [56] and telemedical information systems [33], which could help healthcare providers, expand the scope of their services in the future.

## Improvements to the blockchain architecture on a technical level

The majority of this field's study has concentrated on improving the efficiency of architectures and developed systems by technological improvements for example utilizing smaller data block sizes [74] and reducing transaction propagation delay [68, 74]. Some attention has also been given to problems that have previously been described as possible roadblocks to the successful implementation of blockchain architectures. Memory and CPU specifications [77], as well as accurate node recognition, are among the problems considered in research, grouped under this theme [71]. The efficacy of potential solutions to the above problems has been illustrated in several cases by network and algorithm comparison studies [21, 30, 74]. However, we believe that this theme will continue to progress in the future, necessitating a parallel emphasis on comparative analyses to determine the most powerful networks and algorithms.

## Building predictive capabilities

A similar pattern can be seen in blockchain technology's use in healthcare as it enters the fourth step in development with the rising integration of AI [26]. IoT [50], sensors [47], wireless body area networks [53, 64], big data [49], edge computing [76], and cloud technology have all recently been incorporated through blockchain-based device architecture [59]. Researchers are using such technologies to help them develop systems based on blockchain having predictive capacities for enhancing medical information systems and diagnostics [61, 75]. Prescription fraud avoidance [47], verifiable data generation [77], and automatic claim resolution [47] have all been investigated previously using such frameworks [72]. Furthermore, studies have centered on using blockchain-based technology in supporting providers of health care services with other tasks, e.g data collection on population-level [46] and user identity description [28].

## Enhancement of efficiency

Several researchers in previous literature have attempted to determine how blockchain-based implementation would improve the efficiency of healthcare processes [34, 59, 79, 82]. According to this study, scholars' attention has been drawn to two facets of performance improvement: structures and processes.

## Process

Prior research has focused on improving the efficiency of technological aspects of the processes that are needed to run a blockchain-based healthcare system. Prior research has also focused on developing systems for timely alerts [72] and adverse event reporting [73]. Some research has centered on increasing the computational processes efficiency [57] and thorough evaluation of suggested architectures [60] to ensure that its architecture gives far efficient processing as compared to conventional architectures [71]. Furthermore, studies have proposed changes

in blockchain systems to resolve the alleged risks related to time management, data management, and managing related costs. For example, reviewed research has established mechanisms for reducing the cost of implementation after setting up initially [74], lowering storage costs [65], and making maintenance and storing files of any size is easier [62, 112].

### System

According to our analysis of the current literature, several steps have been applied to enhance the blockchain-based healthcare framework holistically. For example, research has focused on improving system interoperability [27, 49], managing inter-institutional access rights [63, 99], and data management [28, 46, 63, 64, 99]. Enhancing machine scalability [113] and efficiency have also received scholarly attention [60, 65]. Researchers have concentrated their efforts on designing integrated service-oriented architectures [56] and enhancing the generalizability and flexibility of blockchain systems that have been implemented [21, 27, 30].

### Management of data

According to the results, we can conclude that managing the data and medical records is getting more scholarly attention. Existing research has endorsed the blockchain's utilization for medical data management [27, 48, 54, 55, 69, 70, 99, 115]. Furthermore, by integrating heterogeneous forms of data [27, 69, 100], blockchain can aid in the development of an application system of information to manage such PHRs [59, 64]. We define three main aspects of current research in this field based on the SLR.

### Data privacy

Previous research about data security implications of this technology in medical care has focused on handling the privacy of data by maintaining permitted access to the data. According to the study, access control management [76] has gotten a lot of attention [45, 46]. Because of the requirement of protecting the privacy of confidential data by greater transparency, access control, and immutability, this problem is particularly important in healthcare [55]. Prior research has developed a framework based on blockchain to guarantee the delivery of effective services [115], user-centric [114], and access to patient PHRs and other medical data that is safe and encrypted in response to this vital need e.g. [45, 50, 54].

### Data protection

Another main concern discussed in studies on the blockchain aspects of data management in healthcare is the avoidance of unauthorized access and the preservation of data confidentiality to ensure data safety. The majority of the studies that were examined focused on preventing unauthorized access [66] and preventing eavesdropping [71]. Several methods have been proposed to achieve this aim, including efficient authentication [65], biometric authentication [49], user verification [55], and the use of dual signatures [63].

### Data handling

Prior research has addressed the need for legally and legally compliant collection, sharing, and controlling of healthcare data to some extent. According to our findings, few research studies have recognized the importance of monitoring enforcement [100], let alone the criteria and targets for compliance [63, 67, 75]. However, the importance of data integrity has received a lot of attention [56, 69, 70]. Prior research has looked at issues like authentic data mobilization [46, 77], double storage expenditure [68], and eternal data protection [68, 112, 116, 117].

Along with the growing inter-institutional adoption of blockchain, researchers have transformed their attention to the issue of storing and maintaining sensitive data [67] from a variety of sources, including medical devices [52] and health insurance [77]. A few studies have concentrated on the assistance of cross-institutional sharing of data [67], as well as changes in data sharing quality and flexibility [69]. Additionally, previous studies have addressed the need for data processing improvements (e.g. [77]). Some steps for inducing these changes have been suggested in the reviewed studies, such as the successful incorporation of diverse data from various sources of data [99] and the integration of smart contracts [48]. These themes specify that past studies in that area have focused on (i) improving technological features, (ii) managing medical databases, also (iii) identifying unique capabilities in the medical field, where blockchain could make remarkable contributions. Based on emerging trends, it can be said that scholarship in this field is still transforming, with existing facets of healthcare being recognized as possible recipients of using blockchain as a result of technological advancements.

## Research framework for future synthesis

This analysis and review helped in the framework development which was created with the research gaps identified in existing recommendations suggested in previous research. The research model includes 5components that would aid in the development of the healthcare ecosystem based on blockchain, for future research. The research framework for the BCT-based healthcare system is depicted in Fig 10.

**Data sources.** Personal and medical health records are created and managed by the patients using mobile devices, healthcare service suppliers, and pharmaceutics is one of these associated industries., research and insurance [32, 47, 70]. These serve as the foundation of the blockchain architecture and need management by legitimate and regulatory rules. This

### Blockchain-based healthcare system

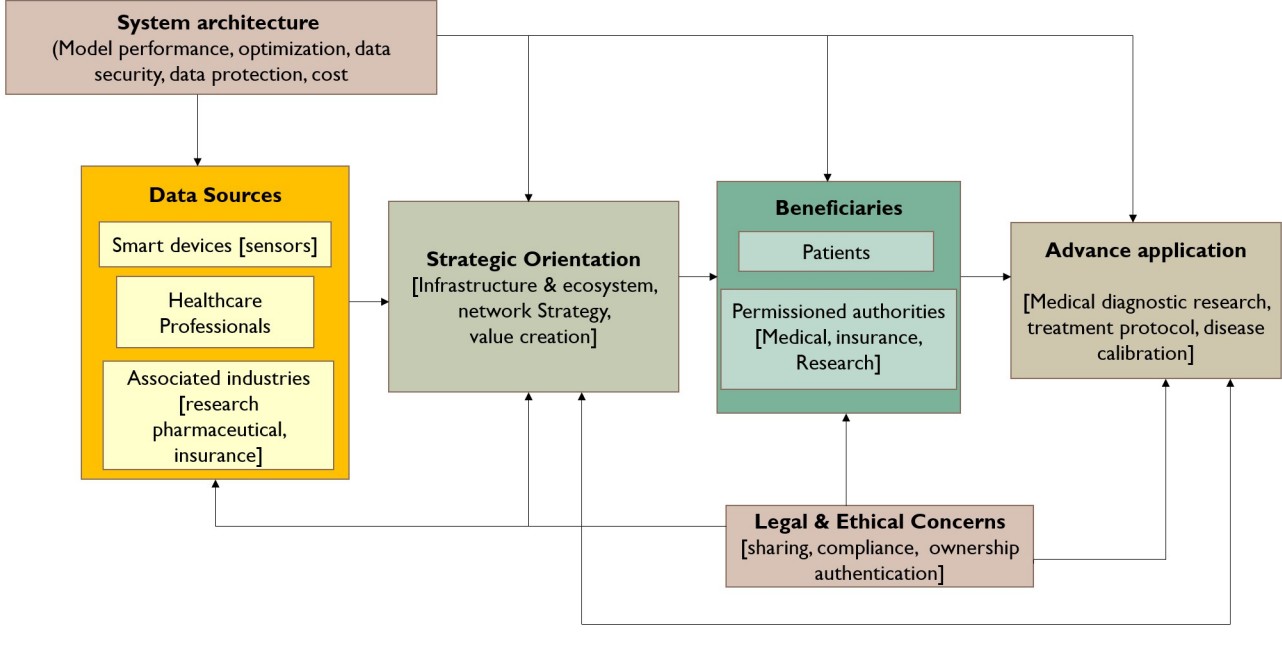

**Fig 10. Research framework.**

technology might aid in the development of authorized databases that information can be retrieved by inter-institutional authorities in collaboration with the required agencies to aid in patient treatment and medical decision-making [21, 30]. Because of the incorporation of newer technologies, such as smart patient tracking devices [53], to increase the comprehensiveness of medical databases, future research should concentrate on handling those data sources.

**System architecture.**   With advances in blockchain technology, the system's architecture will undergo important changes and refinement in terms of the components incorporated into the system of blockchain. Such as using Permissioned consortium blockchain [28] or platforms other than Ethereum [68, 77], could improve current blockchain deployment architectures in the healthcare ecosystem. Also, future research should concentrate on creating techniques for managing system architectures that have been established, particularly the challenging circumstances that can have an impact on the performance and efficiency, for instance, node management [74] and techniques of key distribution [67].

**Blockchain technology strategic implementation.**   In the case of increasing integration of information and communication technology and blockchain over healthcare ecosystems, researchers should have focused on the elements which could impede and assist in the widespread application of blockchain technology. According to our findings, we believe that organizations should think about whether or not, by identifying the key issues throughout this study, blockchain technology might prove a potential source of creating or enhancing value. Strategic problems like resource constraints [50] and technical problems like performance uncertainty [56] and system requirements are these examples [99]. Considering these problems might help scholars to develop blockchain architectures that can offer better functional utility and productivity in terms of resource and output management. This can also guide health system administrators and personnel to adopt a holistic and strategic approach to the potential inclusion of blockchain as an essential component of a company's value chain.

**Beneficiaries.**   Databases built on the blockchain can provide trustworthy information to particular beneficiaries in the sector of healthcare, such as patients who keep ownership of their information. Authorities including doctors, pharmacists, medical researchers, and insurance firms are also beneficiaries. Patients may authorize them to use medical information for a range of purposes, plus collaborative medical decision-making [63], medical informatics and **diagnosis** (S.J. [61]), and fraud prevention [47]. Because of the blurring of the borders between the health system, wellness sectors, and mobile phones, researchers must recognize such beneficiaries ensuring data is accessed by the relevant authority. Moreover, the important thing is maintaining the integrity of data following ethical and legal bindings. As a result, scholars must concentrate to understand the perspective of the user about the perceived advantages and costs of engaging in a blockchain system. It can assist to identify and remove obstacles in the widespread application and use of blockchain.

**Ethical & legal consideration.**   Blockchain applications are addressing critical issues for example authentication, interoperability, and safe sharing of medical data [49, 118–120]. Regardless of the increased emphasis on the blockchain, the acceptance of such concerns may be regarded as a remarkable barrier to its extensive adoption. More emphasis should be placed on regulatory compliance [100] and ethical recommendation for issues like control of ownership and access of patient data [99]. We propose that future scholars take a multidisciplinary approach to determine avenues for resolving ethical and legal compliance issues in multinational or cross-institutional contexts for blockchain adoption. We also argue that there is a need to positively impact the public and appease regulatory agencies by deliberating and highlighting the critical benefits derived by using technology based on blockchain.

## Directions for future research

According to the SLR, we provide a brief summarization of the thematic problems that would require attention from future researchers:

**Deployment of holistic view.**   In case it is critical to find solutions to security and performance-related problems, like interoperability [67] and access-control [68], we argue that scholars must take a broader view of blockchain adoption. This is critical to creating holistic, legally, and ethically compliant [21, 30], robust data management, and authentication procedures in e-health ecosystems [33]. Furthermore [36], argues that context variables like people and culture may play an important role in the development of new technologies. Eventually, we suggest testing blockchain-based electronic health ecosystems in cross-institutional and cross-national contexts to build tailored context-based healthcare solutions in collaborating with different organizations inside the healthcare space, such as research medical centers [60].

**Optimization of the architecture.**   Scholars might focus on improving the efficiency and performance of proposed designs to account for the higher transaction rates which may be expected if blockchain is integrated into healthcare operations in the future [113]. That can be accomplished by dealing with network congestion [69], scalability [99], throughput [76], and bandwidth issues [22].

**Data protection & legal compliance.**   Addressing data, plus user privacy and legal problems will be an important area of future research [21, 30, 53]. These can be directly tackled by designing blockchain protocols in handling healthcare records that can be enforceable by smart-contract [36] and compliant with data and privacy protection regulations, for example, Health Insurance Portability and Accountability Act [31, 36, 53].

**Other technologies integration.**   For improved functionality, deployment of block-chain might be advantageous by the technology with business processes in healthcare [36]. For example, researchers can concentrate to advance the incorporation of edge computing, AI, and ML through blockchain health service ecosystems in developing an improved anticipatory analytic model to provide customized health treatment and diagnostics (e.g. [52, 63, 64]). Furthermore, research may aim to improve accessibility, remote control, and emergency services via the integration of sensors based on IoT. Furthermore, we propose two additional potential directions for future scholars to extend the existing scope of academic boundaries in this sector. First of all, it proposes the requirement to understand the implications of blockchain deployment in more niches in healthcare, but related fields i.e. managing the digital rights of users' [13], drug prescription management [11], and prescription fraud prevention [47]. Furthermore, the research could be conducted to investigate the implications of blockchain usage across the whole health system supply and value chain. It can help scholars better understand user-related interoperability problems and additionally enables creating standard protocols to use systems working under the blockchain.

## Conclusion

This research study is designed to understand completely the application of blockchain in the domain of healthcare. To achieve this goal, SLRs were conducted on nine highly regarded databases using particular protocols to pick out relevant articles for review. The outcomes were used, to sum up, current knowledge on applications of blockchain in the specific sector of medical care, but to also summarize past and the present academic research theme trends in this field. Future research possibilities have been showcased in the form of a synthesized framework created by combining insights from existing restrictions, suggestions, and emerging gaps in current knowledge observed throughout this review.

## Supporting information

**S1 Checklist. PRISMA 2020 checklist.**
(DOCX)

## Author Contributions

**Conceptualization:** Hassaan Malik, Umair Bashir, Aiesha Ahmad, Maheen Ilyas, Muhammad Imran Ali Khan.

**Data curation:** Huma Saeed, Hassaan Malik, Aiesha Ahmad, Shafia Riaz, Wajahat Anwaar Bukhari, Muhammad Imran Ali Khan.

**Formal analysis:** Umair Bashir, Aiesha Ahmad, Maheen Ilyas, Wajahat Anwaar Bukhari.

**Funding acquisition:** Shafia Riaz.

**Investigation:** Aiesha Ahmad.

**Methodology:** Huma Saeed, Aiesha Ahmad, Shafia Riaz, Maheen Ilyas.

**Resources:** Muhammad Imran Ali Khan.

**Supervision:** Hassaan Malik.

**Writing – original draft:** Huma Saeed, Hassaan Malik, Aiesha Ahmad, Maheen Ilyas.

**Writing – review & editing:** Umair Bashir, Wajahat Anwaar Bukhari.

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
