## [Decision Letter · Decision Letter 0]

25 Nov 2021

PONE-D-21-33056Blockchain Technology in Healthcare: A Systematic ReviewPLOS ONE

Dear Dr. Malik,

Thank you for submitting your manuscript to PLOS ONE. After careful consideration, we feel that it has merit but does not fully meet PLOS ONE’s publication criteria as it currently stands. Therefore, we invite you to submit a revised version of the manuscript that addresses the points raised during the review process.

We look forward to receiving your revised manuscript.

Kind regards,

Pandi Vijayakumar, Ph.D

Academic Editor

PLOS ONE

Journal Requirements:

3. We note that Figures 1, 2 and 10 in your submission contain copyrighted images. All PLOS content is published under the Creative Commons Attribution License (CC BY 4.0), which means that the manuscript, images, and Supporting Information files will be freely available online, and any third party is permitted to access, download, copy, distribute, and use these materials in any way, even commercially, with proper attribution. For more information, see our copyright guidelines: http://journals.plos.org/plosone/s/licenses-and-copyright.

a. You may seek permission from the original copyright holder of Figures 1, 2 and 10 to publish the content specifically under the CC BY 4.0 license.

Additional Editor Comments:

Based on the comments of the reviewers, I recommend minor revision for this paper.

Reviewers' comments:

Reviewer's Responses to Questions

**Comments to the Author**

1. Is the manuscript technically sound, and do the data support the conclusions?

Reviewer #1: Yes

Reviewer #2: Yes

2. Has the statistical analysis been performed appropriately and rigorously? 

Reviewer #1: Yes

Reviewer #2: Yes

3. Have the authors made all data underlying the findings in their manuscript fully available?

Reviewer #1: Yes

Reviewer #2: Yes

4. Is the manuscript presented in an intelligible fashion and written in standard English?

Reviewer #1: Yes

Reviewer #2: Yes

5. Review Comments to the Author

Reviewer #1: irst of all, I congratulate all of the authors for working on the significant topic of blockchain technology in health care. I have a few suggestions listed below:

In table 3, write a complete name instead of writing single alphabets.

Correct the sequence of Figure 6. (i.e. Security and Privacy).

Reviewer #2: In this work, the authors have done a survey on Blockchain Technology in Healthcare. The study is compiled by reviewing research articles published in nine well-reputed venues such as IEEE Xplore, ACM Digital Library, Springs Link, Scopus, Taylor & Francis, Science Direct, PsycINFO, Ovid Medline, and MDPI between January 2016to August 2021. A total of 1,192 research studies were identified out of which 51 articles were selected based on inclusion criteria for this SLR that presents the modern information on the recent implications and gaps in the use of BCT for enhancing the healthcare procedures. A framework is developed to address the probable field where future researchers can add considerable value, such as data protection, system architecture, and regulatory compliance. Hence, The paper can be accepted after making the following Minor corrections.

1. There are few grammatical errors in the manuscript. So English proof reading is required. For example, “The research studies of [13] and [14] have been discussed” could be written as “The research studies of [13] and [14] have discussed”.

2. I don’t find X-Axis and Y-axis values in “Figure 3”.

3. Some important recent references are missing, the following references must be totally added in the Section "References" (otherwise, the reference is not enough, then it must be revised again until it is enough):

The Application of the Blockchain Technology in Voting Systems: A Review

Homechain: A blockchain-based secure mutual authentication system for smart homes

Human-in-the-loop-aided privacy-preserving scheme for smart healthcare

Profile Matching for IoMT: A Verifiable Private Set Intersection Scheme

6. PLOS authors have the option to publish the peer review history of their article (what does this mean?). If published, this will include your full peer review and any attached files.

Reviewer #1: No

Reviewer #2: No

---

## [Author Response · Author response to Decision Letter 0]

12 Feb 2022

1) Please explain where the authors obtained the images in Figures 6, 7, 8, 9, and 10 in your submission or if the authors created the image themselves (Please note that we are referring to the actual images within the Figure, rather than the Figure as a whole).

Author response: Thanks for the comment. We have revised the Figures, and all of the Figures are created by the authors of this paper. In addition, Figures 6,7,8,9, and 10 are now Figure 5,6,7,8, and 9 in the updated manuscript.

2) Please state whether the images in Figures 6, 7, 8, 9, and 10 have been previously copyrighted.

Author response: All of the figures are created by the authors. None of the Figures are previously copyrighted. 

3) If any of the images in these figures have been previously copyrighted, we require specific consent from the copyright holder to publish these images in PLOS ONE, under the CC BY 4.0 license. To seek permission from the copyright owner to publish these figures under the Creative Commons Attribution License (CCAL), CC BY 4.0, please contact them with the following text and PLOS ONE Request for Permission form (http://journals.plos.org/plosone/s/file?id=7c09/content-permission-form.pdf):

“I request permission for the open-access journal PLOS ONE to publish XXX under the Creative Commons Attribution License (CCAL) CC BY 4.0 (http://creativecommons.org/licenses/by/4.0/). Please be aware that this license allows unrestricted use and distribution, even commercially, by third parties. Please reply and provide explicit written permission to publish XXX under a CC BY license.”

Please upload the granted permission to the manuscript as a Supporting Information file. In the figure caption of the copyrighted figure, please include the following text: “Republished from [ref] under a CC BY license, with permission from [name of publisher], original copyright [original copyright year].”

Please note that RightsLink permission forms often impose use restrictions that are incompatible with our CC BY 4.0 license, and we are therefore unable to accept these permissions. For this reason, we strongly recommend contacting copyright holders with the PLOS ONE Request for Permission form.

If you are unable to obtain permission from the original copyright holder, please either remove the figure or supply a replacement figure that complies with the CC BY 4.0 license. Please check copyright information on all replacement figures and update the figure caption with source information. If applicable, please specify in the figure caption text when a figure is similar but not identical to the original image used in the study, and is therefore for illustrative purposes only.

Please also clarify if you received explicit written permission from the copyright holders to publish the Cochrane Bias Tool in Figure 4 under CC BY 4.0.

Author response: Figure 4 is deleted from the updated manuscript.

---

## [Decision Letter · Decision Letter 1]

22 Mar 2022

Blockchain Technology in Healthcare: A Systematic Review

PONE-D-21-33056R1

Dear Dr. Malik,

We’re pleased to inform you that your manuscript has been judged scientifically suitable for publication and will be formally accepted for publication once it meets all outstanding technical requirements.

Kind regards,

Pandi Vijayakumar, Ph.D

Academic Editor

PLOS ONE

Additional Editor Comments (optional):

Both the reviewers have given acceptance and hence the paper can be accepted for publication.

Reviewers' comments:

Reviewer's Responses to Questions

**Comments to the Author**

1. If the authors have adequately addressed your comments raised in a previous round of review and you feel that this manuscript is now acceptable for publication, you may indicate that here to bypass the “Comments to the Author” section, enter your conflict of interest statement in the “Confidential to Editor” section, and submit your "Accept" recommendation.

Reviewer #1: All comments have been addressed

Reviewer #2: All comments have been addressed

2. Is the manuscript technically sound, and do the data support the conclusions?

Reviewer #1: Yes

Reviewer #2: Yes

3. Has the statistical analysis been performed appropriately and rigorously? 

Reviewer #1: Yes

Reviewer #2: Yes

4. Have the authors made all data underlying the findings in their manuscript fully available?

Reviewer #1: Yes

Reviewer #2: Yes

5. Is the manuscript presented in an intelligible fashion and written in standard English?

Reviewer #1: Yes

Reviewer #2: Yes

6. Review Comments to the Author

Reviewer #1: author addressed all the review comments satisfactory. Hence, I recommend the acceptance for this paper.

Reviewer #2: The authors have done all the corrections given in the previous round. So the paper can be accepted in the present form.

7. PLOS authors have the option to publish the peer review history of their article (what does this mean?). If published, this will include your full peer review and any attached files.

Reviewer #1: **Yes: **MARIMUTHU KARUPPIAH

Reviewer #2: No

---

## [Editor Report · Acceptance letter]

1 Apr 2022

PONE-D-21-33056R1 

Blockchain Technology in Healthcare: A Systematic Review 

Dear Dr. Malik:

I'm pleased to inform you that your manuscript has been deemed suitable for publication in PLOS ONE. Congratulations! Your manuscript is now with our production department. 

Kind regards, 

on behalf of

Dr. Pandi Vijayakumar 

Academic Editor

PLOS ONE